# Metagenomics Reveal Microbial Effects of Lotus Root–Fish Co-Culture on Nitrogen Cycling in Aquaculture Pond Sediments

**DOI:** 10.3390/microorganisms10091740

**Published:** 2022-08-29

**Authors:** Zhen Yang, Yanhong Yao, Meng Sun, Gu Li, Jianqiang Zhu

**Affiliations:** 1Yangtze River Fisheries Research Institute, China Academy of Fisheries Sciences, Wuhan 430223, China; 2School of Agriculture, Yangtze University, Jingzhou 434025, China

**Keywords:** lotus roots, sediments, nitrogen cycle microorganisms, metabolic pathways

## Abstract

Feed input leads to a large amount of nitrogen-containing sediment accumulating in the substrate in the pond culture process, threatening the safety of aquaculture production. Planting lotus roots (*Nelumbo nucifera* Gaertn.) in ponds can accelerate the removal of bottom nitrogen, while the role of nitrogen cycle-related microorganisms in the removal is still unclear. In this study, eight yellow catfish (*Pelteobagrus fulvidraco*) culture ponds with the same basic situation were divided into fishponds with planted lotus roots and ponds with only fish farming. Sediment samples were taken from the fishponds with planted lotus roots and the ponds with only fish farming before and after fish farming, marked as FPB, FPA, FOB, and FOA, respectively, and subjected to physicochemical and metagenomic sequencing analyses. The results show that the contents of NH_4_^+^, NO_2_^−^, TN, TP, and OM were significantly lower (*p* < 0.05) in FPA than in FOA. The abundance of metabolic pathways for inorganic nitrogen transformation and ammonia assimilation increased considerably after culture compared to the sediments before culture. A total of eight ammonia production pathways and two ammonia utilization pathways were annotated in the sediments of the experimental ponds, with a very high abundance of ammonia assimilation. *Acinetobacter* and *Pseudomonas* (34.67%, 18.02%) were the dominant bacteria in the pond sediments before culture, which changed to *Thiobacillus* (12.16%) after culture. The FPA had significantly higher relative abundances of *Thiobacillus denitrificans* and *Sulfuricella denitrificans*, and the FOA had significantly a higher abundance of *Microcystis aeruginosa* compared to other samples. The massive growth of *Microcystis aeruginosa* provided two new inorganic nitrogen metabolic pathways and one organic nitrogen metabolic pathway for FOA. The relative abundances of these three microorganisms were negatively correlated with NH_4_^+^ content (*p* < 0.01) and significantly positively correlated with AP, OM content, and pH value. Compared with ponds with only fish farming, lotus root–fish co-culture can significantly reduce the nitrogen content in sediment, increase the abundance of denitrifying bacteria, and inhibit algae growth. Still, it has little effect on the abundance of nitrogen cycle-related enzymes and genes. In summary, it is shown that, although lotus roots promote the growth of denitrifying microorganisms in the sediment, nitrogen removal relies mainly on nutrient uptake by lotus roots.

## 1. Introduction

Recently, the aquaculture industry around the world has rapidly grown. In order to obtain high yields to meet the demand for aquatic products, aquaculture activities are increasingly becoming intensive and have high feed inputs [1]. However, recent research showed that only 23.0% of the nitrogen in feed was utilized by aquatic animals [2,3]. At the bottom of ponds, a large quantity of unutilized nitrogen forms sediments. Nitrogen was found to be one of the essential nutrients that give rise to water eutrophication [4], and some nitrogen metabolites, such as ammonia and nitrite, were shown to be bio-toxic to farmed fish [5]. The increase in long-term nitrogen content significantly affects the nitrogen cycle microbial community in sediments [6]. The cultivation of aquatic plants is one of the most commonly used, clean production techniques in pond production. Plants can absorb nitrogen from water and sediments during their growth, improving the substratum environment [7]. Therefore, an in-depth understanding of the effects of plants on nitrogen cycle microorganisms in sediments during the culture process will help to optimize phytoremediation techniques and improve the farming environment, thereby reducing the risk of aquaculture production [8,9]. Currently, research on the application of aquatic plants in ponds focuses on the improvement of technology and the testing of removal effects; an integrated study of the functional potential and functional microbe in nitrogen cycling is missing.

Recently, the polymerase chain reaction–denaturing gradient gel electrophoresis (PCR-DGGE) technique and Illumina Miseq sequencing techniques have been widely used to detect the microbial community structure in aquaculture ponds [10,11]. Although these studies provide information on the structure and functional genes of microbial communities growing in aquaculture ponds, they make it difficult to understand the interactions between different functional microorganisms. The rapid development of the aquaculture system is accompanied by many environmental problems, especially nitrogen pollution [12]. Therefore, many studies have been conducted on nitrogen cycle microgeneration in the aquaculture pond environment. The presence and activity of Ammonia-Oxidizing Bacteria (AOB) have been evaluated in the sediments of four different freshwater aquaculture ponds [13]. Lu et al. [10,14] analyzed the vertical isolation and phylogenetic characteristics of AOB and Ammonia-Oxidizing Archaea (AOA) in the sediments of aquaculture ponds. Dai et al. [15] studied the environmental drivers of AOA and AOB in pond sediments. However, these methods can only illustrate the community structure of environmental microorganisms in the aquaculture pond and cannot reflect the diverse functions in situ. In addition, annotation based on empirical knowledge may be biased due to the uncoupling of microbial community structure and function.

Metagenomic techniques can determine the diversity and functional potential of complex communities. This technique has been applied to investigate the microbial community and metabolic possibilities in the abyssal sea [16], cryospheric ecosystems [17], river [18], lake [19], grassland [20], volcano [21], forest [22], plateau [23], and wetlands [24]. However, the application of metagenomic technology in aquaculture is still scarce.

Studies have shown that aquatic animals utilize only 23.0% of the nitrogen in feed, and the rest is left at the bottom of the pond or discharged into the surrounding water [25]. The transformation of nutrients such as nitrogen and phosphorus in pond-bottom sediments mainly relies on enzymes and microorganisms. This study, based on metagenomics and related functional studies, assembled both phylogenetic and biological functional inventories in ways that can enable community structure and biological processes within fishponds with planted lotus roots. The nitrogen metabolism pathways, relative microbial abundance, and environmental differences were compared between the two groups of pond sediments before and after cultivation. We aimed to investigate the effects of aquaculture activities on nitrogen cycling in sediments and whether planting lotus roots would interfere with nitrogen cycling in sediments.

## 2. Materials and Methods

### 2.1. Experiment Design

Eight identical test ponds were divided into four fishponds with planted lotus roots and four ponds with only fish farming to carry out the yellow jaw fish fingerling culture experiment. The size of each test pond was 20 m × 22 m, and the ecological environment was identical. This study was carried out in the Yaowan Experimental Base of the Yangtze River Fisheries Research Institute of the Chinese Academy of Fishery Sciences in Jingzhou City, Hubei Province (30°16′15″ N, 112°18′31″ E). The base belongs to the subtropical humid monsoon climate, with an average annual temperature of 16.2 °C, an average frost-free period of 252 d, and an average rainfall of 1200 mm. The breeding production time is from April to November every year.

The lotus root cultivar planted in the ponds was Elian No. 5, transplanted on 22 May 2020, with a row spacing of 1.5 m × 2 m (Figure 1). The soil surface of the planting area at the bottom of the pond was kept moist from when the lotus root was planted to when the seeds germinated. After budding, the pond water level gradually increased from 2 up to 50 cm. On 28 July 2020, 12,000 yellow jaw fish/667 m^2^ (0.21 g/tail) were stocked, and artificial pellets were fed to yellow catfish regularly under normal circumstances. The amount provided in the early stage of yellow catfish fry breeding was 5% of the total mass of yellow mulberry fish, 3% in the middle stage, and 2% in the later stage; every morning at 9:00 a.m. and 5:00 p.m., the amount of food was set, accounting for 40% and 60% of the daily amount, respectively.

### 2.2. Sample Sampling and Pre-Treatment

Two sets of sediment samples were collected from the fishponds before planting lotus roots (1 May) and after culture (11 November). The two groups of sediment samples were named FB and FA, respectively. Sediment samples from fishponds with planted lotus roots and ponds with only fish farming collected before cultivation were named FPB and FOB, respectively, and after cultivation were named FPA and FOA. Sediment samples were collected and stored in the following ways: Using a grapple-type sediment collector in each pond’s lotus root planting area, five sediment samples were collected diagonally crossed (five-point method). They were evenly mixed through a 50-mesh sieve, divided into two parts after removing stone and plant residues, and stored in a refrigerator at −20 °C for subsequent physicochemical analysis and microbial determination.

### 2.3. Physicochemical Analysis

Ammonia nitrogen (NH_4_^+^), nitrate nitrogen (NO_3_^−^), nitrite nitrogen (NO_2_^−^), total nitrogen (TN), total phosphorus (TP), organic matter (OM), and dissolved phosphorus (DP) content in the sediments were measured according to the Soil Agrochemical Analysis published by the China Agriculture Press [26]; The pH was measured using a mass ratio of ultrapure water/sediments of 5:1; ultrapure water was added to the sediment, and the pH value was determined after soaking for 1 h.

### 2.4. DNA Extraction and Metagenomic Sequencing

The microbial DNA in the sediments was extracted employing the CTAB method, and the total DNA was tested in quality: 1% agarose gel electrophoresis detected DNA integrity and the ultra-micro spectrophotometer NanoDrop 2000 (Thermo Fisher Scientific, Waltham, MA, USA) detected DNA concentrations. DNA fragments were then terminally repaired, a poly(A) tail was added, sequencing adaptors were removed, and PCR amplification was performed for library construction. The constructed library was quantified using a Qubit 2.0 Fluorometer. The quantification of the libraries was performed with a Qubit 2.0 fluorometer (Qubit), and fragment size profiles were assessed using an Agilent 2100 BioAnalyzer. The Q-PCR method was used to accurately quantify the effective concentration of the library (determined to be > 2 nM) (Shenzhen Microcorn Technology Group Co., Ltd., Shenzhen, China). Employing the high-throughput sequencer Novaseq 6000 (Illumina, San Diego, CA, USA) completed the follow-up metagenomic sequencing and bioinformatics. Each sediment sample had 4 biological replicates with a sequencing depth of 10 GB.

### 2.5. Statistical Analysis

The KneadData software was adopted to control the quality of the original data (based on Trimmomatic), and FastQC was utilized to detect the rationality and effect of the quality control before KneadData and after KneadData [27,28]. Kraken2 and a self-built microbial database (individual microbial databases were downloaded from the Kraken website, merged, and then added to some newly discovered bacterial genome data from the Columbia University lab study) were used to identify the species contained in the samples [29]. Bracken was used to make predictions about the species’ actual relative abundance in the samples [30]. Starting from quality control and reads that remove host genes, HUMAnN2 software (based on DIAMOND) was utilized to filter out reads that failed in comparison (HUMAnN2 default comparison parameters: translated_query_coverage_threshold = 90.0, prescreen_threshold = 0.01, evalue_threshold = 1.0, translated_subject_coverage_threshold = 50.0) [31]. The reads of each sample were then compared to a database (UniRef90) to calculate the relative abundance of UniRef90 proteins (RPKM, reads per kilobase per million, the number of reads (mapped reads), and the abundance after the length of the gene) [32,33]. According to the analysis results of HUMAnN2, the species source of the corresponding function was obtained, and a heat map of the composition of the source of functional species was drawn. The correlation between microbial abundance and environmental factors was analyzed with the Spearman method, and the significance (test of significance) was tested (*p* < 0.05, *p* < 0.01). The graphical drawing was completed using R language and ORIGIN.

## 3. Results

### 3.1. Sediment Physicochemical Properties

The sediment physicochemical properties before and after culture in the two groups of ponds are shown in Table 1. Compared to FOB, FOA had significantly lower NH_4_^+^ and significantly higher (*p* < 0.05) NO_2_^−^, TN, DP, OM, and pH. Compared to FPB, FPA had significantly lower NH_4_^+^ and TP and significantly higher DP, OM, and pH. In FOA, the NH_4_^+^, NO_2_^−^, TN, TP, and OM content were significantly higher than FPA.

### 3.2. Microbial Community Structure

Figure 2 shows the relative abundances of the top 10 microorganisms at the phylum and genus levels in the sediment samples from the experimental ponds. The results indicated that the microbial structure in FA and FB was different. At the phylum level, the dominant phylum in the sediments of the experimental ponds was Proteobacteria, and its relative abundance was 83.92% to 92.67%. After culture, the relative abundances of Actinobacteria, Euryarchaeota, Chloroflexi, Verrucomicrobia, Acidobacteria, and Chlorobi were significantly increased (*p* < 0.05), and Firmicutes and Bacteroidetes were significantly decreased. The microbial community in the sediment changed in different culture patterns. The relative abundance of Proteobacteria (92.67%) in FPA was significantly (*p* < 0.05) higher than that in FOA and FPB (83.92%, 86.77%). The relative abundance of Cyanobacteria in FOA (7.85%) was significantly higher than in FOA and FPA (0.46%, 0.79%).

At the genus level, the dominant genera in FB were *Acinetobacter* and *Pseudomonas* (32.82~36.53%, 17.25~18.79%). Their abundance was significantly decreased (*p* < 0.05) after culture (0–0.03%, 5.2–6.95%). Meanwhile, *Thiobacillus* (1.98~2%), with a lower abundance in FB, was increased dramatically in FA (1.98~2~11.32~13%), leaping into a dominant genus. The relative abundances of *Thiobacillus*, *Dechloromonas*, *Acidovorax**,* and *Burkholderia* significantly increased, and *Acinetobacter* significantly decreased after culture. The relative abundances of *Thiobacillus* and *Dechloromonas* in FPA (13%, 1.83%) were significantly higher than in FOA (11.32%, 2.18%). The relative abundance of *Microcystis* in FOA (6.54%) was significantly higher than in FPA and FOA (0.38%, 0.04%).

### 3.3. Nitrogen Cycling Enzyme Family

The annotation results of UniRef90 were compared with the enzyme database using R programming language to obtain the relative abundance (RPKM) of each metabolic process and functional gene (reads per kilobase per million). Figure 3 shows the thermal images of the enzymes involved in the nitrogen cycle, which were selected according to the literature. The results showed that a total of 19 enzymes were involved in the nitrogen cycle process, of which 9 enzymes were involved in inorganic nitrogen conversion, 7 enzymes were involved in ammonia assimilation, and 3 enzymes were involved in organic nitrogen conversion. There were significant differences in the structure of the enzyme involved in the nitrogen cycle between FA and FB, but there was no significant difference between FOA and FPA. The relative abundances of glutamine synthase, glutamic acid synthase (NADPH), and glutamate synthase (ferredoxin) increased in the sediment after culture, while the relative abundances of nitrate reductase, nitric-oxide reductase (cytochrome c), nitrogenase, and hydroxylamine reductase increased. In addition, nitrate reductase, nitrogenase, and glutamine synthetase increased by more than 50%. This suggested that farming activities may enhance the processes of denitrification, nitrogen fixation, and ammonia assimilation in sediments. None of the sediments in the experimental pond were annotated as ammonia oxidase, but there were plenty of ammonia anabolic-related enzymes. This suggested that ammonia assimilation may be pond sediment’s main ammonia conversion pathway.

### 3.4. Nitrogen Cycle Pathways and Functional Gene Family

To further explore the relationship between nitrogen cycle functional pathways, functional genes, and nitrogen cycling enzymes in the sediments, the annotation results of UniRef90 were compared with the KEGG database (Figure 4 and Figure 5). The path diagrams associated with the nitrogen cycle in the experimental pond annotations (Figure 4) and thermal images (Figure 5) were plotted. The results showed strong denitrification and ammonia assimilation in the sediment. According to the annotation results, it was found that there were four nitrate reduction pathways in the sediment, of which nitrite oxidoreductase and nitrate reductase had higher gene abundances, and their central functional genes were narG, narH, and napA. Figure 3 shows that the experimental pond annotated eight ammonia production pathways. Still, only a few ammonia monooxygenase AmoCABs were present, and genes involved in the complete ammonia oxidation and anaerobic ammonium oxidation and nitrite reduction pathways were not annotated. This suggested that ammonia assimilation was the primary utilization for the large amounts of ammonia in the sediments of the ponds. The abundance of glutamine synthetase in the metabolic processes associated with ammonia assimilation was extremely high, much higher than other metabolic pathways in the nitrogen cycle.

The relative abundances of metabolic pathways related to denitrification, nitrogen fixation, and ammonia assimilation in the sediment increased after culture. The abundances of functional genes such as narG, narH, napA, nirS, norB, nifD, nifH, and hcp involved in inorganic nitrogen metabolism in FA increased by more than 25% compared with those in FB. The abundances of functional genes such as glnA, gltB, gdhA, and nrtC involved in anabolic ammonia metabolism in FA increased by more than 50% compared with that in FB.

Compared with FPA and FOB, FOA added two new inorganic nitrogen metabolic pathways: Ferredoxinitrate reductase (K00367 (narB)) and Ferredoxinitrite reductase (K00366 (nirA)), and an organic nitrogen metabolism pathway: Nitrate transport system NrtD (K15579 (nrtD, cynD)). Glutamate dehydrogenase (K00261 (GLUD1 2, gdhA)) was not annotated in FPA, but was present in other samples.

### 3.5. Gene Abundance of Nitrogen Cycle Microorganisms

According to the analysis results of HUMAnN2 and UniRef90, the species origin and corresponding relative abundance (RPKM) of nitrogen cycle functional genes were obtained. Figure 6 shows the results of the abundances of the three types of nitrogen-metabolizing microorganisms in different metabolic pathways. The results showed that only a small number of microorganisms involved in nitrogen metabolism were annotated. There are eleven microorganisms involved in the metabolic process of ammonia anabolics that can be classified at the species level, of which four were in FOA, two were in FPA, and nine were in FOB and FPB (Figure 6a). A variety of ammonia-assimilating microorganisms were present in FB, but in FB, only *Sulfuricella denitrificans* and *Thiobacillus denitrificans* remained. *Thiobacillus denitrificans*, which has a high abundance of ammonia assimilation genes in FB, was not observed in FA. Compared to other treatments, *Methanosaeta harundinacea* and *Microcystis aeruginosa* were newly added to FOA to participate in ammonia assimilation processes. *Methanosaeta harundinace**a* was involved in the metabolic processes associated with Glutamine synthetase (K01915), and *Microcystis aeruginosa* was involved in the entire glutamate metabolism (K01915, K00266, and K00262) process.

Figure 6b showed the relative abundance of microbial genes involved in organic nitrogen metabolism pathways. The results showed that three and two microorganisms in the sediments in FOA and FPA, respectively, were involved in the metabolic process of organic nitrogen. *Thiobacillus denitrificans* and *Sulfuriricella denitrificans* still existed in FA and had higher gene abundances than in FB, while *Entrerobacter cloacae*, *Thiobacillus denitrificans*, and *Comamonas testosteroni* were not found to be involved in organic nitrogen metabolism in FA. In contrast to other treatments, *Microcystis aeruginosa* was added to the FOA and was involved in metabolic processes related to the Nitrate transport system (15576, K15577, K15578, and K15579).

Figure 6c shows the abundance of microbial genes involved in inorganic nitrogen metabolism pathways. The results showed that three and two microorganisms in the sediments in FOA and FPA, respectively, were involved in the inorganic nitrogen metabolism process. *Thiobacillus denitrificans* and *Sulfuriricella denitrificans* still existed FA. They had higher gene abundances than in FB, while *Ruminococcus torques*, *Enterobacter cloacae*, *Comamonas testosteroni*, and *Pseudodomonas Putida* were not involved in inorganic nitrogen metabolism in FA. In contrast to other treatments, *Microcystis aeruginosa* was added to the FOA and was involved in metabolic processes related to Ferredoxinitrate reductase (K00367), Ferredoxinitrite reductase (K00366), and Hydroxylamine reductase (K05601).

### 3.6. Differences in Microbial Abundance Involved in Nitrogen Cycle

According to Bracken’s results, the relative abundances of nitrogen cycle microorganisms were noted in Figure 6, and a Spearman correlation analysis between their abundances and environmental factors is shown in Figure 7. The results showed that the microorganisms involved in the nitrogen cycle in the sediments of the experimental ponds were mainly *Thiobacillus denitrificans*, *Pseudodomonas Putida*, *Microcystis aeruginosa*, and *Lysinibacillus sphaericu*. Their relative abundances were 1.73~11.81%, 0.75~8.61%, 0.3~5.61%, and 0~1.22%, respectively. The relative abundances of *Thiobacillus denitrificans* and *Sulfuricella denitrificans* increased significantly (*p* < 0.05) in sediments after culture. At the same time, the relative abundances of *Pseudomonas putida*, *Lysinibacillus sphaericus*, *Enterobacter cloacae*, *Comamonas testosteroni*, *Acinetobacter johnsonii*, *Lysinibacillus fusiformis*, and *Methanosaeta harundinacea* all decreased significantly. The relative abundance of *Thiobacillus denitrificans* was negatively correlated (*p* < 0.05) with NH_4_^+^ and TP content in the sediments and was positively correlated (*p* < 0.01) with DP and OM content and pH. The relative abundance of *Pseudomonas putida* was negatively correlated with DP and OM content and pH in the sediments and positively correlated with NH_4_^+^ content.

The relative abundances of *Thiobacillus denitrificans* and *Sulfuricella denitrificans* in FPA were significantly higher (*p* < 0.05) than in FOA. The relative abundance of *Microcystis aeruginosa* in FOA was significantly higher than that in FPA and FOB. Among them, the relative abundances of *Sulfuricella denitrificans* and *Microcystis aeruginosa* were negatively correlated (*p* < 0.01) with NH_4_^+^ content in the sediments and positively correlated (*p* < 0.01) with DP and OM content and pH. In addition, the relative abundance of *Microcystis aeruginosa* was positively correlated (*p* < 0.05). with NO_3_^−^ and TN content in the sediments.

## 4. Discussion

### 4.1. Differences of Nutrient Content and Microbial Abundance in Sediments before and after Culture in Different Culture Models

Aquatic plants, an essential part of the pond ecosystem, affect the water’s redox potential, dissolved oxygen, and pH and play a crucial role in water purification, substrate protection, fish disease prevention, and improving aquaculture efficiency [34]. The results showed that planting lotus roots in the pond could significantly reduce sediment TN, TP, and OM content. Compared with FOA, TN, TP, and OM in FPA decreased by 8.5%, 10.9%, and 8.4%, respectively (Table 1). The decrease in NH_4_^+^ content in sediments is the primary driver of TN degradation, which may be related to the preference of plants for nitrogen absorption. Nitrogen forms in the environment also affect the absorption of nitrogen by plants [35]. In this study, the content of NH_4_^+^ was significantly higher than NO_3_^−^ in pond sediments, and lotus roots may be more inclined to absorb NH_4_^+^ in this environment. The aerobic microbes require oxygen to degrade the OM present in the sediment. Hence, higher OM would consume more oxygen and increase aquaculture risk. Planting lotus roots in ponds can significantly reduce the OM content in sediments and provide a stable environment for pond culture. Our results were similar to Si et al. [36]. This may be related to the ability of plants to promote rhizosphere microbial populations to facilitate the uptake and transformation of soil resources [37].

Compared with FB, the relative abundances of Firmicutes and Bacteroidetes in FA sediments were significantly reduced. The study of Xia et al. showed that these microorganisms often appeared in the intestines of animals [38], while the excrement of fish during the culture process continued to accumulate in the sediments, thereby increasing the abundance of these florae. The abundance of cyanobacteria in FA was significantly increased, which may be related to the feed input during the culture process. Feed inputs bring a large amount of nitrogen and phosphorus nutrients to ponds [39], providing a better propagation environment for algae such as cyanobacteria, resulting in a significant increase in algal microorganisms [40]. In the results at the genus level, we found that the primary algae growing in FOA is the common microcystis (Figure 2b), which is a common planktonic cyanobacterium in ponds and lakes. If it proliferates quickly, it causes lake indigo and produces toxins, which is not conducive to aquaculture [41]. The abundance of *Thiobacillus* in FA was significantly improved and became the dominant genus, and the abundance of *Thiobacillus* in FPA was significantly higher than that in FOA (Figure 2b). Studies have shown that *Thiobacillus* can oxidize hydrogen sulfide produced in sediments into sulfates for absorption and utilization by plants, thus promoting the material circulation and transformation of sulfur elements [42]. The increase in the abundance of *Thiobacillus* in FPA reduces the production of hydrogen sulfide in pond sediments and reduces the risk of culture.

### 4.2. Differences in Nitrogen Cycle-Related Enzymes in Sediments before and after Culture in Different Culture Models

Figure 3 shows a high ammonia production potential in the sediments. Nitrification was essential for reducing ammonia concentrations in intensive culture ponds [43]. In this study, both glutamine synthase and glutamic acid synthase (NADPH) associated with ammonia assimilation were highly abundant (Figure 2), suggesting that ammonium nitrogen in the sediments of the experimental ponds was mainly utilized through ammonia assimilation pathways. The results were similar to Min et al. [44]. This is likely because the high organic content in the sediments inhibits the existing conventional nitrification process [45]. The abundances of nitrate reductase, nitric oxide reductase, nitrogen fixase, hydroxylamine reductase, glutamine synthase, and glutamic acid synthase (NADPH) were significantly increased in FA compared with those in FB (Figure 2). This indicates that the metabolic processes of denitrification, nitrogen fixation, and ammonia assimilation of microorganisms in sediments were enhanced after anthropogenic farming activities. During the culture process, the organic matter content in the sediments significantly increases due to the input of artificial feed (Table 1), and the accumulation of organic matter can enhance the denitrification of the soil [46]. The increased abundance of nitrogenase and hydroxylamine reductase genes in sediments enhanced ammonia production potential. The production of a large amount of ammonia in the sediments promoted the further enrichment of genes related to ammonia assimilation. Between the two culture models, there was less difference in enzyme abundance associated with nitrogen cycling between FA and FB (Figure 2). This result shows that the lotus root does not promote or inhibit the nitrogen cycle processes in pond sediments. Therefore, the significant reduction in nitrogen content in FPA sediments compared with FOA may mainly rely on the nutrient uptake and rhizosphere microbial transformation of the lotus root itself [47,48].

### 4.3. Differences in Nitrogen Cycle-Related Metabolic Pathways in the Sediments before and after Culture in Different Culture Models

Previous studies have shown an inseparable relationship between gene abundance associated with nitrogen conversion and nitrogen conversion efficiency [49]. In this study, six metabolic pathways associated with nitrate reduction were noted in the pond sediments, among which the abundances of narG, narH, and napA genes were higher (RPKM > 100) (Figure 4). This indicates that the pond sediments may have a strong nitrate reduction process. Our results show that there are lower levels of NO_3_^−^ (Table 1) in pond sediments. NarG and narH, as membrane-bound nitrate reductase, are extremely sensitive to oxygen content and play an important role in denitrification in anaerobic and hypoxic environments [50]. In the later stages of farming, the water depth gradually increases, and the oxygen content in the sediments gradually decreases. The addition of nitrogen-rich substances such as culture feed and fish manure create a good environment for the growth of narG- and narH-related microorganisms. Studies have shown that napA can be used as a basis for aerobic denitrification [51], and the abundance of the napA gene increases significantly in the later stages of culture. It showed that adding exogenous organic matter promotes the sediment’s anaerobic and aerobic denitrification process. The increase in the abundances of nirS and norB genes in the post-culture sediments ensures the smooth progress of the subsequent process of denitrification. Compared with FOB and FPA, ferrotropinitric acid and nitrite reductase-led denitrification pathways appear in FOA. Flores et al. [52] suggested that photosynthetic nitrate assimilation is possible in some algae, and the enzymes involved in the process of photosynthetic nitrate assimilation are ferrotropin-dependent nitrate reductase and nitrite reductase. The abundance of algae in FAO was significantly higher than that in FOB and FPL (Figure 1), indicating that the increase in this part of the algae adds a new path for the denitrification process.

### 4.4. Differences in Nitrogen Cycle-Related Microorganism in the Sediments before and after Culture in Different Culture Models

After culture, the relative abundances of *Thiobacillus denitrificans* and *Sulfuricella denitrificans* increased significantly (*p* < 0.05). In sediments, *Thiobacillus denitrificans* are a dominant species involved in ammonia assimilation and organic and inorganic nitrogen metabolism pathways. This shows that *Thiobacillus denitrificans* play a vital role in the nitrogen cycle of sediments. The continuous introduction of high-protein exogenous feeds in culture systems has increased the OM content in sediments (Table 1). Nevertheless, studies have shown that adding organic matter facilitates the conversion of sulfur, thus contributing to the increase in *Thiobacillus denitrificans* [53]. This was further demonstrated by the positive correlation between Thiobacillus denitrifying and OM content in sediments (Figure 7). *Thiobacillus denitrificans* occupy a very high relative abundance in post-culture sediments, and they can oxidize sulfides by using nitrates as electron receptors in anaerobic environments [54]. Only very low abundances of nitrification-related metabolic pathways were annotated in post-culture sediments. Still, metabolic pathways associated with nitrate conversion in organic states began to be annotated (Figure 5), suggesting that the nitrate nitrogen utilized in *Thiobacillus denitrificans* in post-culture sediments was likely derived from the decomposition of organic nitrogen. The nitrogen content in the sediments of the ponds studied was very rich, which provided favorable conditions for the growth of algal microorganisms; therefore, the abundance of *Microcystis aeruginosa* in FPA increased significantly. While *Microcystis aeruginosa* adds new nitrogen metabolism to sediments, it dramatically increases the risk of cyanobacterial blooms in culture ponds [55]. However, there were no significant differences in *Microcystis aeruginosa* before and after culture in the sediments of the lotus pond (Figure 7), indicating that the cultivation of lotus roots can effectively inhibit the growth of *Microcystis aeruginosa* in the sediments. This may be explained by the fact that lotus roots compete with algae for nutrition and sunlight and that higher plants produce an algae-allelopathy effect, which inhibits algae growth [56].

## 5. Conclusions

The sediments in the experimental ponds had processes of strong denitrification, nitrogen fixation, and ammonia assimilation, and the culture process would further increase the abundances of enzymes and functional genes of their associated metabolic pathways. There are multiple ammonia production pathways in sediments, with ammonia assimilation being the primary ammonia utilization pathway rather than nitrification. Planting lotus roots in ponds can significantly reduce the content of nitrogen and phosphorus nutrients in sediments and inhibit the growth of *Microcystis aeruginosa*, but they have little impact on single-cycle-related enzymes and metabolic pathways. After culture, fish-only ponds had increased *Microcystis aeruginosa* abundance responses in sediments. However, the increase in *Microcystis aeruginosa* provided new denitrification pathways. The central nitrogen circulating microorganisms in FB and FA were *Pseudomonas putida* and *Thiobacillus denitrificans*, respectively. Culturing lotus roots promotes the increase in the abundance of *Thiobacillus denitrificans* in the sediments.

## Figures and Tables

**Figure 1 microorganisms-10-01740-f001:**
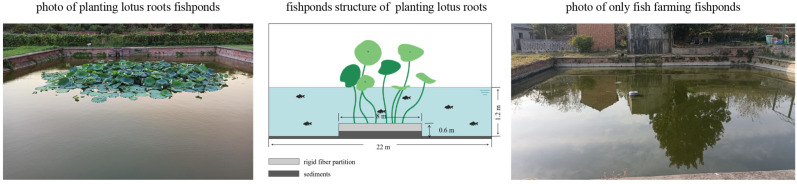
Planting lotus root group (**left**), lotus ridge structure (**middle**), and only fish farming group (**right**) in yellow catfish culture experimental ponds. To prevent lotus roots from overgrowing the whole pond, the sediment around the PL pond was gathered towards the middle to form an area for planting lotus roots. A rigid fiber partition was used to enclose the root planting area and was inserted into the silt for 0.60 m. The shape surrounded by the partition was circular and had a diameter of 8 m.

**Figure 2 microorganisms-10-01740-f002:**
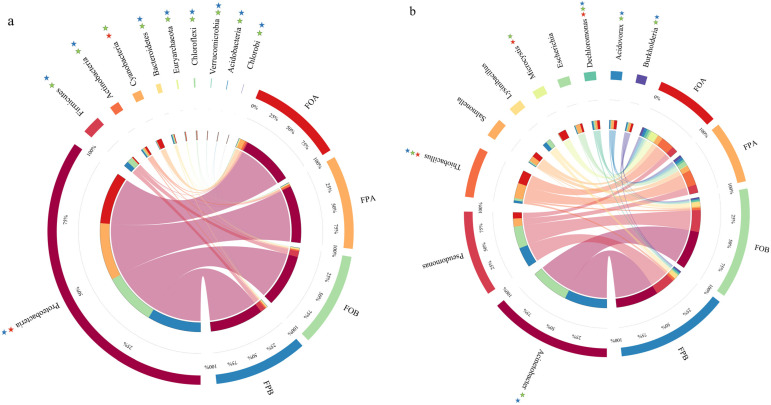
Relative abundances of dominant microorganism phyla (**a**) and genera (**b**) in sediment. FPB: Sediment samples in fishponds with planted lotus roots collected before cultivation; FOB: Sediment samples in ponds with only fish farming collected before cultivation; FPA: Sediment samples in fishponds with planted lotus roots collected after cultivation; FOA: Sediment samples in ponds with only fish farming collected after cultivation. Red, Green, and Blue stars indicate that the differences between FOA and FPA, FOA and FOB, FPA and FPB are significant, respectively; *p* < 0.05, *n* = 4.

**Figure 3 microorganisms-10-01740-f003:**
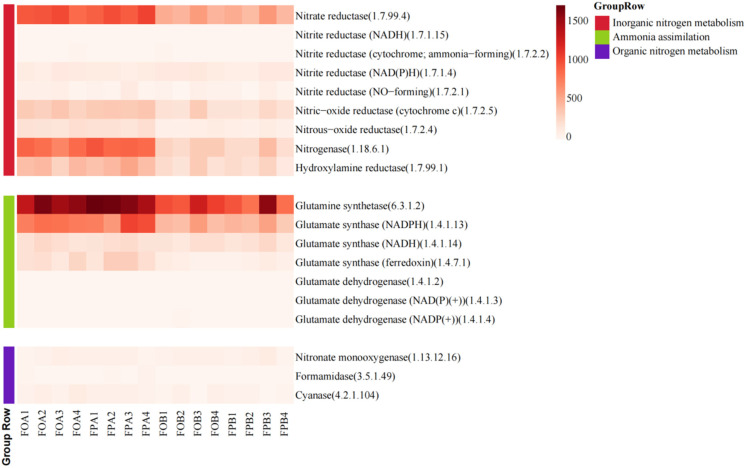
The heat map shows the proportion and abundances of enzyme families involved in the nitrogen cycle in the ponds’ sediments. The results of enzyme abundance in the figure were obtained by comparison with the ENZYME database. The number behind the enzyme is the EC number of the enzyme. The same is true below.

**Figure 4 microorganisms-10-01740-f004:**
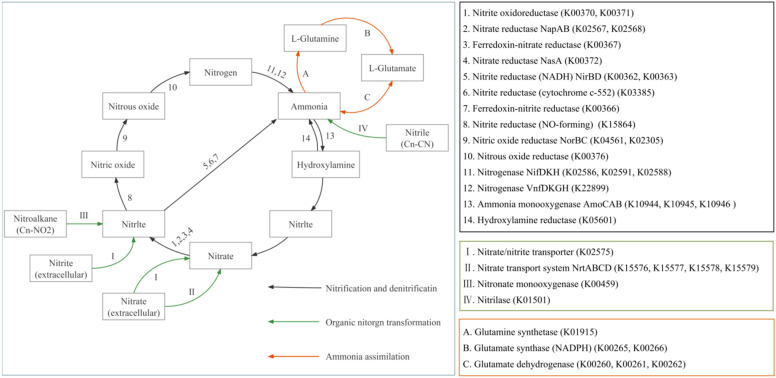
Map of the nitrogen cycling pathways and processes annotated in the pond sediments.

**Figure 5 microorganisms-10-01740-f005:**
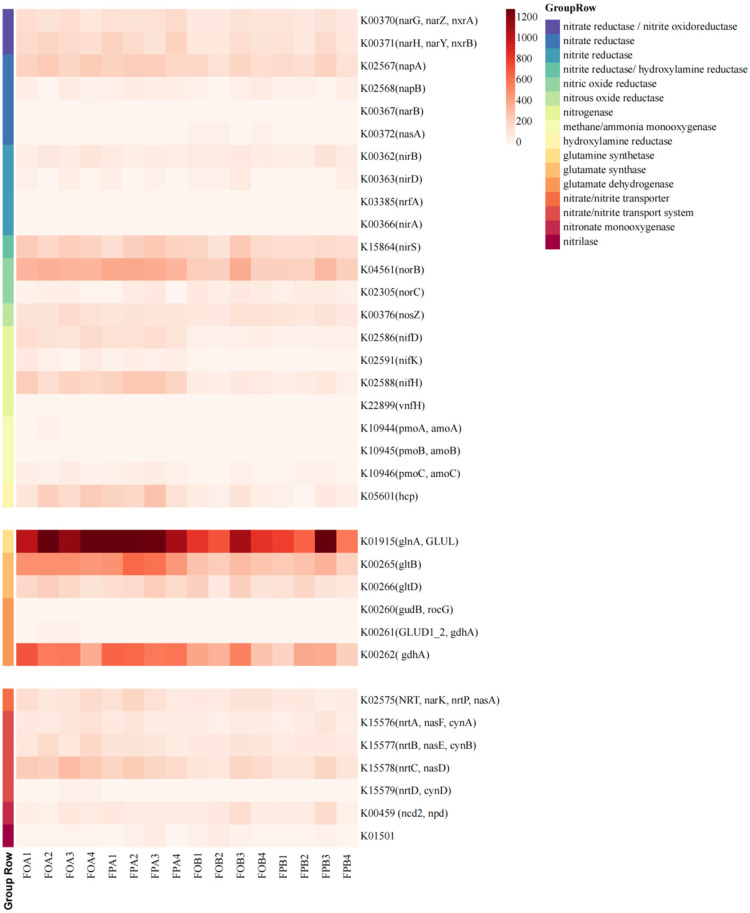
The heat map shows the proportion and abundance of functional pathways and gene families involved in the nitrogen cycle in pond sediments. The numbers on the ordinate represent the nitrogen cycle metabolic pathways annotated by KEGG, and the contents in brackets are the functional genes involved in the metabolic pathway.

**Figure 6 microorganisms-10-01740-f006:**
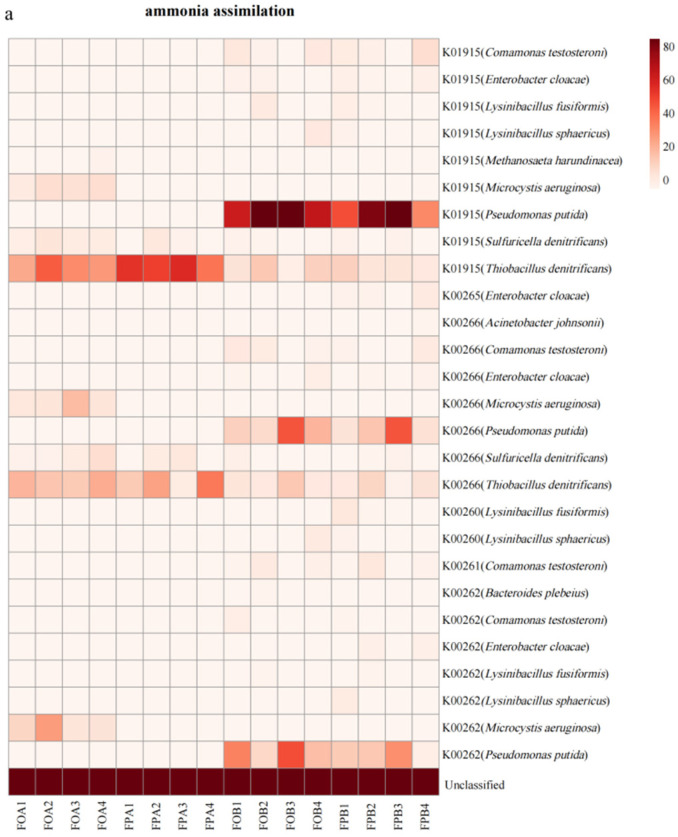
The heat map shows the abundance of genes in different functional pathways of microorganisms that were annotated to the nitrogen cycle gene family. The numbers on the ordinate represent the nitrogen cycle metabolic pathways annotated by KEGG, and the contents in brackets are the microorganisms involved in the metabolic pathway.

**Figure 7 microorganisms-10-01740-f007:**
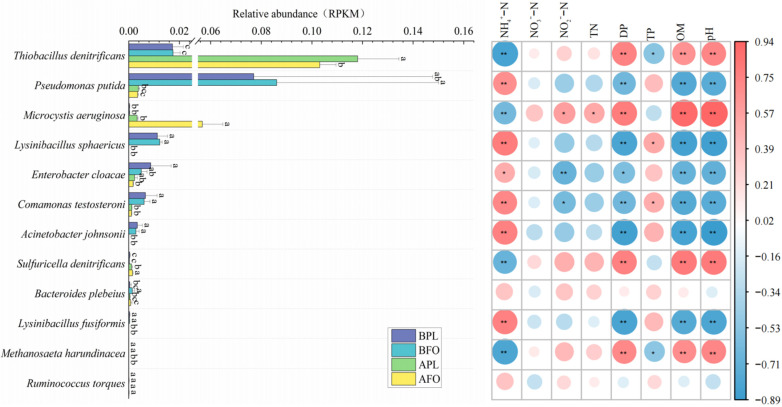
The relative abundances of microorganisms annotated to the nitrogen cycle gene family. Different letters designate remarkable differences between different samples (*p* < 0.05, LSD test, *n* = 4). The heat map on the right indicates the correlation between microbial abundance and environmental factors. The size of the circle indicates the magnitude of the correlation coefficient, and the color indicates its positive or negative correlation. ** represents significant correlation at 0.01 level, *n* = 16; * represents significant correlation at 0.05 level, *n* = 16.

**Table 1 microorganisms-10-01740-t001:** Physicochemical factors of pond sediments before and after culture in different culture modes.

Sample	NH_4_^+^ (mg kg^−1^)	NO3− (mg kg^−1^)	NO_2_^−^ (mg kg^−1^)	TN (g kg^−1^)	DP (mg kg^−1^)	TP (g kg^−1^)	OM (g kg^−1^)	pH
FOA	147.07 ± 13.35 b	13.27 ± 2.76 a	1.31 ± 0.19 a	2.28 ± 0.06 a	36.47 ± 2.61 a	1.38 ± 0.07 a	70.94 ± 0.94 a	7.88 ± 0.03 a
FPA	67.48 ± 7.07 c	11.04 ± 0.70 a	0.62 ± 0.09 b	2.09 ± 0.08 b	38.67 ± 6.70 a	1.23 ± 0.07 b	64.99 ± 1.39 b	7.81 ± 0.06 a
FOB	176.34 ± 7.89 a	12.47 ± 2.98 a	0.65 ± 0.07 b	2.12 ± 0.05 b	17.08 ± 1.97 b	1.43 ± 0.03 a	61.73 ± 0.64 c	7.66 ± 0.06 b
FPB	167.98 ± 3.76 a	9.62 ± 2.83 a	0.60 ± 0.07 b	2.16 ± 0.05 b	14.22 ± 1.66 b	1.36 ± 0.06 a	61.31 ± 1.66 c	7.64 ± 0.05 b

**Note:** Data followed by lowercase letters in each column are separated by one-way ANOVA (LSD test, *p* = 0.05, *n* = 4). FPB: Sediment samples in fishponds with planted lotus roots collected before cultivation; FOB: Sediment samples in ponds with only fish farming collected before cultivation; FPA: Sediment samples in fishponds with planted lotus roots collected after cultivation; FOA: Sediment samples in ponds with only fish farming collected after cultivation. NO_3_^−^: nitrate nitrogen; NH_4_^+^: ammonium nitrogen; NO_2_^−^: nitrite nitrogen; TN: total nitrogen; DP: dissolved phosphorus; TP: total phosphorus; OM: organic matter. The concentrations of all indexes are those of dry matter.

## Data Availability

The data that support the findings of this study are available from the corresponding author upon reasonable request.

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
