# Peer review of "Metagenomics Reveal Microbial Effects of Lotus Root–Fish Co-Culture on Nitrogen Cycling in Aquaculture Pond Sediments"

_microorganisms, 2022, doi:10.3390/microorganisms10091740_

Round 1

Reviewer 1 Report

Dear Authors,

The subject of the study is interesting and topical, with scientific and practical importance.

The introduction is presented correctly, in accordance with the subject. Numerous scientific articles, in concordance to the topic of the study, were consulted.

Methodology of the study was clearly presented, and appropriate to the proposed objectives.

The obtained results are important and have been analyzed and interpreted correctly, in accordance with the current methodology.

The discussions are appropriate, in the context of the results, and was conducted compared to other studies in the field.

The scientific literature, to which the reporting was made, is recent and representative in the field.

Some suggestions and corrections were made in the article.

The following aspects are brought to the attention of the authors.

1.

Authors name and Affiliation

It is recommended to follow the Instructions for Authors and Microsoft Word template, Microorganisms journal

eg

Zhen Yang1,2, Meng Sun1, Gu Li2* and Jianqiang Zhu1*” instead of “Zhen Yang1,2, Meng Sun1, Gu Li2* and Jianqiang Zhu1*

2.

“(Nelumbo nucifera Gaertn)” instead of “(Nelumbo nucifera Gaertn)”

3.

It is recommended to use the Equation Editor to write the ionic forms correctly.

It is “NH+” and not “4+

It is “NO-” and not “2-

It is recommended to check in several places in the article and correct it, if necessary.

4.

Italic Font style for species name

It is also recommended to check the name of some species.

Thiobacillus denitrificans” instead of “Thiobacillus denitrificans”

Sulfuricella denitrificans” instead of “Sulfuricella denitrificans”

Methanosaeta harundinacea” instead of “Methanosaeta harundinacea”

Microcystis aeruginosa” instead of “Microcystis aeruginosa”

Enterobacter cloacae” instead of “Entrerobacter cloacae”

Pseudomonas putida” instead of “Pseudomonas putida”

Comamonas testosteroni” instead of “Comamonas testosteroni”

Microcystis aeruginosa” instead of “Microcystis aeruginosa”

Ruminococcus torques” instead of “Ruminococcus torques”

Enterobacter cloacae” instead of “Enterobacter cloacae”

Comamonas testosteroni” instead of “Comamonas testosteroni”

Pseudomonas putida” instead of “Pseudodomonas Putida”

Lysinibacillus sphaericus” instead of “Lysinibacillus sphaericu”

Acinetobacter johnsonii” instead of “Acinetobacter johnsonii”

Lysinibacillus fusiformis” instead of “Lysinibacillus fusiformis”

Methanosaeta harundinacea” instead of “Methanosaeta harundinacea”

5.

It is recommended to check the subtitle "3.2. Figures, Tables and Schemes", page 6, row 297.

According to the Instructions for Authors, and Microsoft Word template Microorganisms journal, it is recommended that "Figures should be placed in the main text near to the first time they are cited."

6.

Table 1 can be expanded horizontally to present the content correctly

7.

To present "Author Contributions", it is recommended to follow the Instructions for Authors, and Microsoft Word template, Microorganisms journal.

eg

The following statements should be used “Conceptualization, X.X. and Y.Y.; methodology, X.X.;

8.

References

The entire References chapter needs to be revised.

It is recommended that you check the text settings in the References chapter to be according to Instructions for Authors, and Microsoft Word template, Microorganisms journal.

eg

“Cripps, S.J.; Bergheim, A. Solids management and removal for intensive land-based aquaculture production systems. Aquac. Eng. 2000, 22(1-2), 33-56. Doi. https://doi.org/10.1016/S0144-8609(00)00031-5”

Instead of

“Cripps, S.J. and A. Bergheim, Solids management and removal for intensive land-based aquaculture production systems. Aquacultural engineering, 2000. 22(1-2): p. 33-56.”

“Jin, J.; Wang, Y.; Wu, Z.; Herhazy, A.; Lan, J.; Zhao, L.; Liu, X.; Chen, N.; Lin, L. Transcriptomic analysis of liver from grass carp (Ctenopharyngodon idellus) exposed to high environmental ammonia reveals the activation of antioxidant and apoptosis pathways. Fish Shellfish Immunol. 2017, 63, 444-451. Doi: 10.1016/j.fsi.2017.02.037”

instead of

“Jin, J., et al., Transcriptomic analysis of liver from grass carp (Ctenopharyngodon idellus) exposed to high environmental ammonia reveals the activation of antioxidant and apoptosis pathways. Fish & Shellfish Immunology, 2017. 63: p. 444-451.”

Author Response

Comments and Suggestions for Authors

Dear Editors and Reviewers:

Thank you very much for your letter and for the reviewers’ comments on our manuscript entitled “Metagenomic reveals microbial effects of louts root-fish co-culture on nitrogen cycling in aquaculture pond sediments”(ID:microorganisms-1847669). Those comments are all valuable and very helpful for revising and improving our paper, as well as the important guiding significance to our researches. We have substantially revised our manuscript providing details in the current version. We have studied the comments carefully and the amendments are highlighted in red in the revised manuscript. We also submitted a revised manuscript with the changes highlighted in yellow. We hope that the revision is acceptable and look forward to hearing from you soon.

With best wishes.

Sincerely Yours,

Zhen Yang

Address correspondence to: Yangtze River Fisheries Research Institute, Chinese Academy of Fishery Sciences, Wuhan 430223, China.

Below, please find the comments in black, followed by our responses in red. Exact changes in the manuscript are also presented in red font. 

Answers to reviewers:

Reviewer #1:

Specific Comments:

The subject of the study is interesting and topical, with scientific and practical importance.

The introduction is presented correctly, in accordance with the subject. Numerous scientific articles, in concordance to the topic of the study, were consulted.

Methodology of the study was clearly presented, and appropriate to the proposed objectives.

The obtained results are important and have been analyzed and interpreted correctly, in accordance with the current methodology.

The discussions are appropriate, in the context of the results, and was conducted compared to other studies in the field.

The scientific literature, to which the reporting was made, is recent and representative in the field.

Some suggestions and corrections were made in the article.

The following aspects are brought to the attention of the authors.

Comment 1: Authors name and Affiliation

It is recommended to follow the Instructions for Authors and Microsoft Word template, Microorganisms journal.

“Zhen Yang1,2, Meng Sun1, Gu Li2*and Jianqiang Zhu1*” instead of “Zhen Yang1,2, Meng Sun1, Gu Li2* and Jianqiang Zhu1*

Response 1: We thank the reviewer for this insightful comment. We have revised the entire References chapter according to Instructions for Authors, and Microsoft Word template, Microorganisms journal. As shown below:

Zhen Yang1,2, Meng Sun1, Gu Li2* and Jianqiang Zhu1  (Page 1, lines 4)

Comment 2: “(Nelumbo nucifera Gaertn)” instead of “(Nelumbo nucifera Gaertn)”

Response 2: Thanks very much for pointing out my mistake. We have replaced Nelumbo nucifera Gaertn to Nelumbo nucifera Gaertn. As shown below:

(Nelumbo nucifera Gaertn) (Page 1, lines 12)

Comment 3: It is recommended to use the Equation Editor to write the ionic forms correctly.

It is “NH+” and not “4+”

It is “NO-” and not “2-”

It is recommended to check in several places in the article and correct it, if necessary.

Response 3: Thank you for your good suggestion. We have checked and modified the ionic forms correctly by using Equation Editor, but there was a problem. In this study, we determined nitrate nitrogen and nitrite nitrogen, respectively. If NO- was used to represent nitrate nitrogen, it is challenging to represent nitrite nitrogen. However, in your suggestion, we found that the expressions of NH4+-N, NO3--N,and NO2--N were problematic. Therefore, we changed NH4+-N, NO3--N,and NO2--N to NH4+, NO3-, and NO2-, respectively. As shown below:

NH4+ (Page 1, lines 19, 30; Page 3, lines 133; Page 4, lines 178, 181, 183; Page 5, lines 189; Page 11, lines 337, 340 ......)

NO3- (Page 3, lines 133; Page 4, lines 183; Page 5, lines 189; Page 11, lines 347; Page 12, lines 368 ......)

NO2- (Page 1, lines 19; Page 4, lines 179, 181, 183; Page 5, lines 189; Page 5, lines 189)

Comment 4: Italic Font style for species name

It is also recommended to check the name of some species.

Thiobacillus denitrificans” instead of “Thiobacillus denitrificans”

Sulfuricella denitrificans” instead of “Sulfuricella denitrificans”

Methanosaeta harundinacea” instead of “Methanosaeta harundinacea”

Microcystis aeruginosa” instead of “Microcystis aeruginosa”

Enterobacter cloacae” instead of “Entrerobacter cloacae”

Pseudomonas putida” instead of “Pseudomonas putida”

Comamonas testosteroni” instead of “Comamonas testosteroni”

Microcystis aeruginosa” instead of “Microcystis aeruginosa”

Ruminococcus torques” instead of “Ruminococcus torques”

Enterobacter cloacae” instead of “Enterobacter cloacae”

Comamonas testosteroni” instead of “Comamonas testosteroni”

Pseudomonas putida” instead of “Pseudodomonas Putida”

Lysinibacillus sphaericus” instead of “Lysinibacillus sphaericu”

Acinetobacter johnsonii” instead of “Acinetobacter johnsonii”

Lysinibacillus fusiformis” instead of “Lysinibacillus fusiformis”

Methanosaeta harundinacea” instead of “Methanosaeta harundinacea”

Response 4: Thank you for your good suggestion. We have modified the species name to italic font style. As shown below:

Thiobacillus denitrificans (Page 1, lines 26; Page 10, lines 292-341; Page 14, lines 449-464; Page 15, lines 487)

Sulfuricella denitrificans (Page 1, lines 26; Page 10, lines 291; Page 11, lines 332-344; Page 14, lines 449)

Methanosaeta harundinacea (Page 1, lines 27; Page 10, lines 294, 314; Page 11, lines 330 ......)

Microcystis aeruginosa (Page 1, lines 27-29; Page 10, lines 294, 305, 314; Page 11, lines 330 ......)

Enterobacter cloacae (Page 1, lines 27-29; Page 10, lines 294, 305, 314; Page 11, lines 330 ......)

Pseudomonas putida (Page 10, lines 312; Page 11, lines 334)

Comamonas testosteroni (Page 10, lines 304, 313; Page 11, lines 334)

Microcystis aeruginosa (Page 1, lines 27; Page 10, lines 294, 297, 305, 314; Page 11, lines 330, 343 ......)

Ruminococcus torques (Page 10, lines 312)

Enterobacter cloacae (Page 10, lines 312; Page 11, lines 334)

Comamonas testosteroni (Page 10, lines 304, 313; Page 11, lines 334)

Pseudomonas putida (Page 10, lines 292, 313; Page 11, lines 334, 340; Page 15, lines 486)

Lysinibacillus sphaericus (Page 11, lines 334)

Acinetobacter johnsonii (Page 11, lines 335)

Lysinibacillus fusiformis (Page 11, lines 335)

Methanosaeta harundinacea (Page 10, lines 294, 295; Page 11, lines 335)

Comment 5: It is recommended to check the subtitle "3.2. Figures, Tables and Schemes", page 6, row 297.

According to the Instructions for Authors, and Microsoft Word template Microorganisms journal, it is recommended that "Figures should be placed in the main text near to the first time they are cited."

Response 5: Thank you for your good suggestion. We have checked the Figures, Tables and Schemes", and page 6 row 297 in section 3.2 , etc, and placed Figures in the main text near to the first time they are cited. As shown below:

Figure 1. Planting Lotus root group (left) and lotus ridge structure (middle) and only fish farming group (right) in yellow catfish culture experimental ponds. To prevent lotus roots from overgrowing the whole pond, the sediment around the PL pond was gathered towards the middle to form an area for planting lotus roots. A rigid fiber partition was used to enclose the root planting area and was inserted into the silt for 0.60 m. The shape surrounded  by the partition was circular and had a diameter of 8 m. (Page 3, lines 112-119)

Figure 2. Relative abundances of dominant microorganism phyla (a) and genus (b) in sediment. FPB: Sediment samples in fishponds of planting lotus roots collected before cultivation; FOB: Sediment samples in only fish farming ponds collected before cultivation; FPA: Sediment samples in fishponds of planting lotus roots collected after cultivation; FOA: Sediment samples in only fish farming ponds collected after cultivation. Red, Green, and Blue stars indicate the differences between FOA and FPA, FOA and FOB, FPA and FPB are significant, respectively; P < 0.05, n=4. (Page 6, lines 215-220)

Figure 3. The heat map shows the proportion and abundance of enzyme families involved in the nitrogen cycle in the ponds sediments. The results of enzyme abundance in the figure were obtained by comparison with the ENZYME database. The number behind the enzyme is the EC number of the enzyme, The same below. (Page 7, lines 241-245)

Figure 4. Map of the nitrogen cycling pathways and processes annotated in the pond sediments. (Page 8, lines 275-276)

Figure 5. The heat map shows the proportion and abundance of functional pathways and gene families involved in the nitrogen cycle in pond sediments. The numbers on the ordinate represent the nitrogen cycle metabolic pathways annotated by KEGG, and the contents in brackets are the functional genes involved in the metabolic pathway. (Page 9, lines 277-281)

Figure 6. The heat map shows the abundance of genes in different functional pathways of microorganisms that were annotated to the nitrogen cycle gene family. The numbers on the ordinate represent the nitrogen cycle metabolic pathways annotated by KEGG, and the contents in brackets are the microorganisms involved in the metabolic pathway. (Page 11, lines 318-323)

Figure 7. The relative abundance of microorganisms annotated to the nitrogen cycle gene family. Different letters designate remarkable differences between different sample(P < 0.05, LSD test, n=4). The heat map on the right indicates the correlation between microbial abundance and environmental factors. The size of the circle indicates the magnitude of the correlation coefficient, and the color indicates its positive or negative correlation. ** represents significant correlation at 0.01 level, n=16; * represents significant correlation at 0.05 level, n=16. (Page 12, lines 349-355)

Comment 6: Table 1 can be expanded horizontally to present the content correctly

Response 2: Thank you for your good suggestion. We have modified Table 1 according to your suggestion. As shown below:

Table 1. Physicochemical factors of pond sediments before and after culture in different culture modes.

Sample

NH4+

(mg kg-1)

NO3-

(mg kg-1)

NO2-

(mg kg-1)

TN

(g kg-1)

DP

(mg kg-1)

TP

(g kg-1)

OM

(g kg-1)

pH

FOA

147.07±13.35 b

13.27±2.76 a

1.31±0.19 a

2.28±0.06 a

36.47±2.61 a

1.38±0.07 a

70.94±0.94 a

7.88±0.03 a

FPA

67.48±7.07 c

11.04±0.70 a

0.62±0.09 b

2.09±0.08 b

38.67±6.70 a

1.23±0.07 b

64.99±1.39 b

7.81±0.06 a

FOB

176.34±7.89 a

12.47±2.98 a

0.65±0.07 b

2.12±0.05 b

17.08±1.97 b

1.43±0.03 a

61.73±0.64 c

7.66±0.06 b

FPB

167.98±3.76 a

9.62±2.83 a

0.60±0.07 b

2.16±0.05 b

14.22±1.66 b

1.36±0.06 a

61.31±1.66 c

7.64±0.05 b

Note: Data followed with lower letters in each column are separated by one-way ANOVA (LSD test, P = 0.05, n=4). FPB: Sediment samples in fishponds of planting lotus roots collected before cultivation; FOB: Sediment samples in only fish farming ponds collected before cultivation; FPA: Sediment samples in fishponds of planting lotus roots collected after cultivation; FOA: Sediment samples in only fish farming ponds collected after cultivation. NO3-: nitrate nitrogen. NH4+: ammonium nitrogen; NO2-: nitrite nitrogen; TN: total nitrogen; DP: dissolved phosphorus; TP: total phosphorus; OM: organic matter. The concentrations of all indexes are those of dry matter (Page 4-5, lines 182-191)

Comment 7: To present "Author Contributions", it is recommended to follow the Instructions for Authors, and Microsoft Word template, Microorganisms journal.eg

“The following statements should be used “Conceptualization, X.X. and Y.Y.; methodology, X.X.;”

Response 7: Thank you for your good suggestion. We have presented "Author Contributions" following the instructions for Authors, and Microsoft Word template, Microorganisms journal. As shown below:

Writing—original draft preparation, review and editing, Zhen Yang; Data curation and investigation, Meng Sun; Supervision and project administration, Jianqiang Zhu; Resources and project administration, Gu Li. All authors have read and agreed to the published version of the manuscript.” Please turn to the CRediT taxonomy for the term explanation. (Page 15, lines 489-492)

Comment 8: References

The entire References chapter needs to be revised.

It is recommended that you check the text settings in the References chapter to be according to Instructions for Authors, and Microsoft Word template, Microorganisms journal.

eg

“Cripps, S.J.; Bergheim, A. Solids management and removal for intensive land-based aquaculture production systems. Aquac. Eng. 2000, 22(1-2), 33-56. Doi. https://doi.org/10.1016/S0144-8609(00)00031-5”

Instead of

“Cripps, S.J. and A. Bergheim, Solids management and removal for intensive land-based aquaculture production systems. Aquacultural engineering, 2000. 22(1-2): p. 33-56.”

“Jin, J.; Wang, Y.; Wu, Z.; Herhazy, A.; Lan, J.; Zhao, L.; Liu, X.; Chen, N.; Lin, L. Transcriptomic analysis of liver from grass carp (Ctenopharyngodon idellus) exposed to high environmental ammonia reveals the activation of antioxidant and apoptosis pathways. Fish Shellfish Immunol. 2017, 63, 444-451. Doi: 10.1016/j.fsi.2017.02.037”

instead of

“Jin, J., et al., Transcriptomic analysis of liver from grass carp (Ctenopharyngodon idellus) exposed to high environmental ammonia reveals the activation of antioxidant and apoptosis pathways. Fish & Shellfish Immunology, 2017. 63: p. 444-451.”

Response 8: Thank you for your good suggestion. We have modified the authors name and affiliation according to the instructions for authors and microsoft word template, microorganisms journal. As shown below:

References

  1. Cripps, S.J.; Bergheim, A. Solids management and removal for intensive land-based aquaculture production systems. Aquacultural engineering 2000, 22, 33-56. Doi:  https://doi.org/10.1111/j.1461-0248.2007.01139.x
  2. Gondwe, M.J.; Guildford, S.J.; Hecky, R.E. Tracing the flux of aquaculture-derived organic wastes in the southeast arm of Lake Malawi using carbon and nitrogen stable isotopes. Aquaculture 2012, 350, 8-18. Doi: https://doi.org/10.1016/j.aquaculture.2012.04.030
  3. Lananan, F.; Hamid, S.H.A.; Din, W.N.S.; Khatoon, H.; Jusoh, A.; Endut, A. Symbiotic bioremediation of aquaculture wastewater in reducing ammonia and phosphorus utilizing Effective Microorganism (EM-1) and microalgae (Chlorella sp.). International Biodeterioration & Biodegradation 2014, 95, 127-134. Doi: https://doi.org/10.1016/j.ibiod.2014.06.013
  4. Conley, D.J.; Paerl, H.W.; Howarth, R.W.; Boesch, D.F.; Seitzinger, S.P.; Havens, K.E.; Lancelot, C.; Likens, G.E. Controlling eutrophication: nitrogen and phosphorus. 2009, 323, 1014-1015. Doi: https://doi.org/10.1126/science.11​​67755
  5. Jin, J.; Wang, Y.; Wu, Z.; Hergazy, A.; Lan, J.; Zhao, L.; Liu, X.; Chen, N.; Lin, L. Transcriptomic analysis of liver from grass carp (Ctenopharyngodon idellus) exposed to high environmental ammonia reveals the activation of antioxidant and apoptosis pathways. Fish & Shellfish Immunology 2017, 63, 444-451. Doi: https://doi.org/10.1016/j.fsi.2017.02.037
  6. Sun, R.; Wang, F.; Hu, C.; Liu, B. Metagenomics reveals taxon-specific responses of the nitrogen-cycling microbial community to long-term nitrogen fertilization. Soil Biology and Biochemistry 2021, 156, 108214. Doi: https://doi.org/10.1016/j.soilbio.2021.108214
  7. Chen, C.; Zhao, T.; Liu, R.; Luo, L. Performance of five plant species in removal of nitrogen and phosphorus from an experimental phytoremediation system in the Ningxia irrigation area. Environmental Monitoring and Assessment 2017, 189, 1-13. Doi: https://doi.org/10.1007/s10661-017-6213-y

…… (Page 15-19, lines 505-655)

Reviewer 2 Report

The work is interesting, but the experimental part is not explained well enough. Therefore, it is necessary to correct the experimental part. Microorganisms are written in italics. Some claims in the discussion are not correct, please correct the above.

Author Response

Dear reviewer

Thanks for your positive and constructive comments and suggestions on our manuscript. It is very important for the improvement of quality to this paper. We have replied to your comments and suggestions point by point. We believe with the revised paper would be fit to publish in The Microorganisms journal.

Reviewer #2:

Specific Comments:

The work is interesting, but the experimental part is not explained well enough. Therefore, it is necessary to correct the experimental part. Microorganisms are written in italics. Some claims in the discussion are not correct, please correct the above.

Comment 1: How did You isolated and identified these bacteria? In the experimental section this is not explained.

Response 1: We thank the reviewer for this insightful comment. In this study, the methods of screening nitrogen cycling microorganisms were described in different chapters. Respectively in 2.5. Statistical Analysis (Page 4, lines 165-171), 3.4 Nitrogen cycle pathways and functional gene family, and 3.5. Gene abundance of nitrogen cycle microorganisms. Firstly, we obtained the metabolic pathways related to nitrogen cycling through the existing literature. (Page 7, lines 247-249) Then, according to the analysis results of HUMAnN2, the species source of the corresponding function was obtained. (Page 9, lines 283-285) Finally, Bracken's results obtained the proportion of nitrogen-cycling microorganisms in the total microorganisms. (Page 11, lines 325-327)

Comment 2: Did You determined nitrite?

Response 2: Thank you for your question. We did this experiment when we started the project, but we forgot to point it out in the test method. We’re sorry that we did not declare it in the text. This has been supplemented in the article. As shown below:

Ammonia nitrogen (NH4+), nitrate nitrogen (NO3-), nitrite nitrogen (NO2-), total nitrogen (TN), total phosphorus (TP), organic matter (OM), and dissolved phosphorus (DP) content in the sediments were measured according to the Soil Agrochemical Analysis published by China Agriculture Press. (Page 3, lines 133-136)

Comment 3: Did You determined TOC/TC and IC?

Response 3: Thank you for your insightful question. At the beginning of the experiment,TOC/TC and IC in the sediments has not been carried out, but we discussed about it in the future work section and wish to finish it in our continued study.

Comment 4: Did You determined the total number of bacteria present in the sediment, CFU?

Response 4: The reviewer has made a very good point here. The suggested by the review is interesting and would provide valuable information to the study. But we have not determined the total number of Bacteria Present in the test. Therefore, relative abundance was used in the comparison of microbial differences in this paper.

Comment 5: What is this?‘DP’

Response 5: We are sorry that we did not expressed well and made you confused. We determined the dissolved phosphorus (DP) content when we started the project, but the relevant description was forgotten in the section of materials and methods. Thank you very much for your comment on this issue. We changed our Methods as bellows:

Ammonia nitrogen (NH4+), nitrate nitrogen (NO3-), nitrite nitrogen (NO2-), total nitrogen (TN), total phosphorus (TP), organic matter (OM), and dissolved phosphorus (DP) content in the sediments were measured according to the Soil Agrochemical Analysis published by China Agriculture Press. (Page 3, lines 133-136)

Comment 6: Italic

Response 6: We appreciated the reviewer’s attention to detail, and we have corrected the text as suggested.

Thiobacillus denitrificans (Page 1, lines 26; Page 10, lines 292-341; Page 14, lines 449-464; Page 15, lines 487)

Sulfuricella denitrificans (Page 1, lines 26; Page 10, lines 291; Page 11, lines 332-344; Page 14, lines 449)

Methanosaeta harundinacea (Page 1, lines 27; Page 10, lines 294, 314; Page 11, lines 330 ......)

Microcystis aeruginosa (Page 1, lines 27-29; Page 10, lines 294, 305, 314; Page 11, lines 330 ......)

Enterobacter cloacae (Page 1, lines 27-29; Page 10, lines 294, 305, 314; Page 11, lines 330 ......)

Pseudomonas putida (Page 10, lines 312; Page 11, lines 334)

Comamonas testosteroni (Page 10, lines 304, 313; Page 11, lines 334)

Microcystis aeruginosa (Page 1, lines 27; Page 10, lines 294, 297, 305, 314; Page 11, lines 330, 343 ......)

Ruminococcus torques (Page 10, lines 312)

Enterobacter cloacae (Page 10, lines 312; Page 11, lines 334)

Comamonas testosteroni (Page 10, lines 304, 313; Page 11, lines 334)

Pseudomonas putida (Page 10, lines 292, 313; Page 11, lines 334, 340; Page 15, lines 486)

Lysinibacillus sphaericus (Page 11, lines 334)

Acinetobacter johnsonii (Page 11, lines 335)

Lysinibacillus fusiformis (Page 11, lines 335)

Methanosaeta harundinacea (Page 10, lines 294, 295; Page 11, lines 335)

Comment 7: Why is this in separate section? Please incorporate these in section results

Response 7: Thank you for your insightful question. We downloaded the proposed template of the journal when we submitted the manuscript, but we misunderstood the placement of the chart positions in the template. Thank you very much for pointing out this problem, which effectively improves the integrity of the structure of the article. We have now placed the chart closest to the first reference.

Comment 8: dry matter or volatile matter? Please add in subscript on which basis are all these values

Response 8: We thank the reviewer for this insightful comment. All the sediment physicochemical indices present in this study were concentration at dry matter, and we have corrected the text as suggested. As shown below:

Table 1. Physicochemical factors of pond sediments before and after culture in different culture modes.

Sample

NH4+

(mg kg-1)

NO3-

(mg kg-1)

NO2-

(mg kg-1)

TN

(g kg-1)

DP

(mg kg-1)

TP

(g kg-1)

OM

(g kg-1)

pH

FOA

147.07±13.35 b

13.27±2.76 a

1.31±0.19 a

2.28±0.06 a

36.47±2.61 a

1.38±0.07 a

70.94±0.94 a

7.88±0.03 a

FPA

67.48±7.07 c

11.04±0.70 a

0.62±0.09 b

2.09±0.08 b

38.67±6.70 a

1.23±0.07 b

64.99±1.39 b

7.81±0.06 a

FOB

176.34±7.89 a

12.47±2.98 a

0.65±0.07 b

2.12±0.05 b

17.08±1.97 b

1.43±0.03 a

61.73±0.64 c

7.66±0.06 b

FPB

167.98±3.76 a

9.62±2.83 a

0.60±0.07 b

2.16±0.05 b

14.22±1.66 b

1.36±0.06 a

61.31±1.66 c

7.64±0.05 b

Note: Data followed with lower letters in each column are separated by one-way ANOVA (LSD test, P = 0.05, n=4). FPB: Sediment samples in fishponds of planting lotus roots collected before cultivation; FOB: Sediment samples in only fish farming ponds collected before cultivation; FPA: Sediment samples in fishponds of planting lotus roots collected after cultivation; FOA: Sediment samples in only fish farming ponds collected after cultivation. NO3-: nitrate nitrogen. NH4+: ammonium nitrogen; NO2-: nitrite nitrogen; TN: total nitrogen; DP: dissolved phosphorus; TP: total phosphorus; OM: organic matter. The concentrations of all indexes are those of dry matter (Page 4, lines 182-191)

Comment 9: This is not well written. Oxygen will not be used by organic matter, but by organisms that are using organic matter

Response 9: Thank you for your insightful question. We are sorry that we did not expressed well and made you confused. We changed our Methods as bellows:

The aerobic microbes require oxygen to degrade the OM present in the sediment. Hence, higher OM would consume more oxygen and increase aquaculture risk. (Page 12, lines 369-371)

Round 2

Reviewer 2 Report

The paper can be published in this form.